# Shape-Memory Polymers Hallmarks and Their Biomedical Applications in the Form of Nanofibers

**DOI:** 10.3390/ijms23031290

**Published:** 2022-01-24

**Authors:** Silvia Pisani, Ida Genta, Tiziana Modena, Rossella Dorati, Marco Benazzo, Bice Conti

**Affiliations:** 1Otorhinolaryngology Unit, Department of Surgical Sciences, Fondazione IRCCS Policlinico San Matteo, 27100 Pavia, Italy or silvia.pisani01@universitadipavia.it (S.P.); m.benazzo@smatteo.pv.it (M.B.); 2Department of Drug Sciences, University of Pavia, 27100 Pavia, Italy; ida.genta@unipv.it (I.G.); tiziana.modena@unipv.it (T.M.); rossella.dorati@unipv.it (R.D.)

**Keywords:** Shape-Memory Polymers (SMPs), electrospinning, shape-memory polymers nanofibers (SMPNs), engineered scaffold, biomedical applications

## Abstract

Shape-Memory Polymers (SMPs) are considered a kind of smart material able to modify size, shape, stiffness and strain in response to different external (heat, electric and magnetic field, water or light) stimuli including the physiologic ones such as pH, body temperature and ions concentration. The ability of SMPs is to memorize their original shape before triggered exposure and after deformation, in the absence of the stimulus, and to recover their original shape without any help. SMPs nanofibers (SMPNs) have been increasingly investigated for biomedical applications due to nanofiber’s favorable properties such as high surface area per volume unit, high porosity, small diameter, low density, desirable fiber orientation and nanoarchitecture mimicking native Extra Cellular Matrix (ECM). This review focuses on the main properties of SMPs, their classification and shape-memory effects. Moreover, advantages in the use of SMPNs and different biomedical application fields are reported and discussed.

## 1. Shape-Memory Polymers

Shape-Memory Polymers (SMPs) are considered a kind of smart material able to modify size, shapes, stiffness or strain in response to different external (heat, electric and magnetic field, water or light) stimuli including physiological ones such as pH, body temperature and ion concentration [1]. The ability of SMPs is to memorize the original structure shape before triggered exposure and after deformation, in the absence of the stimulus, and to recover without any help their original shape. SMPs have shown excellent features such as multishape-memory effect, easy processing and large deformation. For these reasons, they have attracted great attention allowing them to be used in a broad range of applications including sensors, smart textiles, aerospace, robotics, biomedicine, and so on [2,3]. In particular, biodegradable SMPs (BSMPs) have found great application in the medical field due to their absence of toxicity and their ability in facilitating minimally invasive procedures. Moreover, the main aspects to be considered during the development of biodegradable SMPs for biomedical applications are the trigger mechanism for shape transformation, biocompatibility, sterilization, shape-memory performance, degradation products’ toxicity, and mechanical properties [4]. The most common biodegradable SMPs used in biomedical fields are reported in Table 1.

The chemical architectures common to all SMPs is the possession of molecular switching segments (sensible to a stimulus) and net-points. Molecular switching segments are considered the “soft” portion acting as reversible phases, and net-points are the hardest part that determine the fixed shape [20].

Crystalline, liquid crystalline and amorphous phases are introduced as switch units and are responsible for shape fixity and recovery when an external stimulus is applied. Conversely, crystalline phases, chemical crosslinks, chain entanglements, interpenetrating networks and/or interlocked supramolecular complexes are introduced as net-points to stabilize the fixed shapes.

There are many different ways to classify SMPs depending on crosslinking type, shape-memory effect, macroscopic form and triggering stimulus. One classification divides SMPs into physically crosslinked and chemically crosslinked. Physically crosslinked SMPs possess a network made by noncovalent bonds, and in chemically crosslinked SMPs there are covalent bonds. SMPs can be processed in different types of structures and can be classified as shape-memory blocks, shape-memory foams, shape-memory fibers, and shape-memory films [21]. SMPs can be also classified according to their shape-memory effect (SME), as one-way shape-memory effect (OWSME), two-way reversible shape-memory effect (TWSME) and multiple-SME (Figure 1), whose definitions are reported here below.
OWSMEs: materials lose shape reversibility meaning that when SMP recovers its original shape, another step is necessary to induce temporary shape.TWSMEs: are materials able to switch between original and temporary shape several times without applying a further reshaping. TWSMEs can also be called reversible shape-memory effect (reversible SME), and the polymers with reversible SME are called reversible SMPs.Multiple-SMEs: materials that show two or more than two temporary shapes in addition to original shape. Transition from first to second temporary shape is allowed by external stimulus and further stimulation permits them to return to the polymer original shape.

Behl et al. developed fully reversible bidirectional SMPs using the strategy to assign two functions to two separate structural units, which are linked to each other at molecular level. The crystallizable domains associated with higher T_m_, SGD (shifting-geometry determining domains), determined the shape-shifting geometry, while the domains associated with lower T_m_, AD (actuator domains), were responsible for shape-memory effect activation. They used a copolyester urethane network of poly(ω-pentadecalactone) and poly(ε-caprolactone)(PPD-PCL 25:75 wt%), where PPD acts as the geometry-determining domains (T_m_, SGD) while PCL provided the actuator domains (T_m_, AD). Original Shape A (bow) was obtained after programming by deforming shape B (helix) at T_reset_ (100 °C), cooling to T_low_ and subsequent heating to T_high_. The reversible SME occurred as reversible shifts between shape A at T_high_ and shape B at T_low_ (Figure 2). The shape shifts occurred at the same temperature independent from the shape shifting geometry [22,23,24,25,26,27].

An example of multiple-SME is Nafion, a commercial thermoplastic polymer, used as SMPs in bulk film form. Electrospun Nafion nanofibers (ENNMs) showed some advantages compared to films, such as large specific surface area, high aspect ratio (length to diameter ratio), strong and mutual penetration of other substances, fine fabric structure, and high porosity (better adsorption and filtration properties). ENNMs possess a broad T_g_ range (60–170 °C) and are able to memorize more than two shapes, specifically they are defined as quintuple-shape-memory membranes due to their ability to memorize five different shapes. The shape recovery can be triggered by heat in a single-, triple-, quadruple-, and quintuple-shape-memory cycle [28].

Considering the type of stimulus able to induce a shape-memory effect, it is possible to differentiate between chemically induced and physically induced SMPs. More innovative SMPs can be stimulated by two or more than two different stimuli and are called multistimuli-responsive SMPs [29,30]. Different stimuli-induced SMPs are schematized in Figure 3 and explained in the following subsections.

### 1.1. Physically Induced SMPs

SMPs that are induced by external physical triggers such as temperature, UV light and electric and magnetic field are classified as physically induced SMPs.

#### 1.1.1. Thermally Induced SMPs

The most studied type of SMPs are thermally induced SMPs. These SMPs can be activated by direct thermal application, and when the temperature applied is higher than polymer transition temperature (T_trans_), a transitory shape can be programmed.

Thermally induced SMPs’ mechanism is based on two thermal transitions: (1) melting temperature (T_m_-based) and (2) glass transition temperature (T_g_-based).

Thermally induced SMPs’ mechanism based on T_m_ is typical of chemically and physically crosslinked polymers. This type of SMPs possess as a switching segment a multiblock copolymer with a low melting phase, while the permanent network is represented by the copolymers’ high melting temperature phase. The most common T_m_-based SMPs are polyolefins, polyethers and polyesters, showing a low melting temperature soft phase, and a crystalline hard phase that remains unchanged at high temperature. For these materials, switching temperature depends on branching degree and crosslinking density.

SMPs whose T_g_ is above 25 °C can be used as T_g_-based materials. Due to their broader glass transition interval, T_g_-based polymers show a slow shape recovery compared to T_m_-based SMPs. However, slow shape recovery and T_g_ near to the physiological body temperature are attractive properties for biomedical applications because a slow shape recovery is preferred not only for some special clinical purposes, such as orthodontic applications, but also for avoiding insertion-induced tissue damage [20,21].

Thermally T_g_-based SMPs are sensitive to temperature changes due to their characteristic glass transition temperature (T_g_), above which they are in a rubbery state and can be deformed to a secondary shape from a primary shape by application of an external stress. The induced deformed shape can then be fixed by cooling the polymer below its T_g_; afterwards, the primary shape can be recovered by heating the polymer above its T_g_ [31].

Thermally responsive SMPs can be activated by direct or indirect contact with heat sources. Combining SMPs with other responsive materials (nanofillers) such as Fe_3_O_4_, gold and silver nanoparticles (AuNPs and AgNPs), carbon nanotubes (CNTs), graphene oxide (GO) and cellulose nanocrystals, makes it possible to use remote temperature control by magnetic and electric field, microwaves, UV (ultraviolet) and NIR (near infrared) irradiations [32,33]. In these systems, heating generation is induced by molecular vibration, and energy magnitude is directly proportional to the nanofillers’ concentration and particle size.

At the moment, the only thermoresponsive device based on SMP in clinical trial is TrelliX Embolic Coil System (NCT03988062), a medical device used in embolization of medium-to-large, ruptured or unruptured cerebral aneurysms. The TrelliX Embolic Coil is a stretch-resistant platinum–tungsten alloy coil augmented with a self-expanding porous SMP. Upon deployment in the target lesion and exposure to an aqueous environment and body temperature, the porous SMP will slowly self-expand. TrelliX Embolic Coil devices are covered by the following issued patents (US8133256 and US10010327) [34,35,36].

#### 1.1.2. Light-Induced SMPs

Light-induced shape-memory polymers are obtained by incorporation of photosensitive units, such as azobenzenes with photoisomerizable property, triphenylmethane leuco derivatives and cinnamate group with photoreversibility, into the polymeric structure, which is responsible for their permanent shape, resulting in a photoresponsive network [37]. Fixation of the temporary shape is achieved by application of an external stress and irradiation with UV light at a wavelength lower than a predetermined wavelength (nm). To induce polymer shape recovery, irradiation at a wavelength higher than the predetermined wavelength is sufficient, resulting in the cleavage of photosensitive crosslinks.

Lendlein et al. reported that grafted polymers containing cinnamic groups can be deformed and fixed into predetermined shapes such as (but not exclusively) elongated films and tubes, arches or spirals, by UV light illumination. New shapes are stable for long time laps and can recover their original shape at room temperature when exposed to UV light of a different wavelength [38].

#### 1.1.3. Electric/Magnetic-Induced SMPs

Electric/magnetic-induced SMPs can also be called indirect thermal-induced SMPs because the stimuli can achieve the remotely controlled shape-memory effect and because electricity or magnetism are finally transformed to Joule heating in order to endow shape recovery.

Shape recovery of electric-induced SMPs is stimulated by heat generated by electricity. When a certain voltage is applied, due to the Joule heating effect, the electric energy is converted into thermal energy that, overcoming the transition temperature, activates SMPs [39]. Thermal conductivity of electrically induced SMPs is a very important factor because good thermal conductivity helps electrically induced SMPs to reach the shape-memory transition temperature in a short time, which is conducive to improving the recovery speed of electrically induced SMPs.

Moreover, SMPs can be divided into thermoplastic SMPs and thermoset SMPs. Thermoset SMPs have higher stiffness and dimensional stability and have better environmental durability compared to thermoplastic SMPs. Epoxy resins are widely used thermosetting plastics with many excellent properties: high mechanical strength and thermal stability, good resistance to acid and alkali corrosion, and good formability. Wang et al., developed a silver-plated chopped carbon fiber (Ag/CCF) filled into hydrogenated bisphenol A epoxy resin (H-EP) to fabricate the electroinduced shape-memory polymer composites. When the content of Ag/CCF is higher than 1.8 wt%, the Ag/CCF/H-EP composites exhibit excellent electroactive shape-memory performance, and the shape recovery rate of the composites is more than 92% [40].

Crosslinked polycaprolactone (PCL) was implemented with multiwalled carbon nanotube (MWCNT) nanocomposite and demonstrated ability to recover its permanent shape from the deformed shape in 120 s under 60 V voltage [41].

### 1.2. Chemically Induced SMPs

Shape memory polymers induced by chemical triggers such as pH, water, solvents or biological agents are classified as chemically induced SMPs.

#### 1.2.1. pH-Induced SMPs

Physiological human body pH varies depending on human body area and on pathological conditions. Therefore, pH-sensitive SMPs find an interesting application in the biomedical field [42].

These types of SMPs are obtained by addition of reversible crosslinks or sensitive pH groups (amino, carboxyl and sulfonic groups) that, when exposed to different pH environments, cause inclusion or complexes’ dissociation.

For example, Hane et al., obtained a pH SMP prepared by cross-linking β-cyclodextrin-modified alginate (β-CD-Alg) and diethylenetriamine-modified alginate (DETA-Alg). The pH-reversible β-CD-DETA-inclusion complexes serve as a reversible phase, and the cross-linked alginate chains serve as a fixing phase. When the complex is immersed in a solution at pH 11.5, is possible to induce a temporary shape, and the original shape is recovered upon pH shift to 7. The recovery ratio and the fixity ratio were 95.7 ± 0.9% and 94.8 ± 1.1%, respectively. Because the shape transition pH value is quite close to the physiological body pH, this material is convenient and safe to be used with high potential for medical applications [43].

Li et al., obtained a pH-responsive SMP blending poly (ethylene glycol)–poly(ε-caprolactone)-based polyurethane with functionalized cellulose nanocrystals (CNCs). CNCs, functionalized with pyridine moieties (CNC–C_6_H_4_NO_2_), which act as switch units in the polymer matrix to achieve shape-memory behavior; at a high pH value, the CNC–C_6_H_4_NO_2_ had attractive interactions regarding hydrogen bonding between pyridine groups and hydroxyl moieties, while at a low pH value, the interactions reduced or disappeared due to the protonation of pyridine groups, which are a Lewis base [44].

#### 1.2.2. Water-Induced SMPs

Water is the most common and nontoxic fluid suitable for biomedical applications. Bonded water can affect SMPs hydrogen bonds and increase chains’ flexibility, also causing T_g_ reduction. For example, water as a plasticizer can reduce the transition temperature of crosslinked polyvinylalcohol (PVA), and allow original shape recovery at lower temperature [45].

Torbati et al. developed a light-emitting SMPs web composed of poly(vinyl acetate) (PVAc) combined with indocyanine green (a dye emitting in the NIR region). PVAc-webs were able to shrink when immersed in water at 25 °C and 50 °C, and showed significantly higher fluorescence intensity compared to the films. They conclude that dye-containing webs that feature water-triggered contraction could be used as medical devices, such as feeding tubes or catheters, internal suturing, antimicrobial medical devices, packaging, drug delivery, and temperature sensors [46].

Cellulose nanowhiskers (CNW), CNCs and microcrystalline cellulose (MCC) are sensitive to water and their inclusion into a polymer can generate water-sensitive SMPs. Examples were reported in which CNW and MCC were introduced in polyurethane and poly (D,L-lactide) matrices in order to achieve a faster switchable shape-memory effect at physiological temperature (37 °C) [47,48].

#### 1.2.3. Enzymatically Triggered SMPs

Some SMPs could be triggered directly by biological activity possessing cytocompatible shape-memory-triggering mechanisms. Enzymatically sensitive polymers are often naturally occurring biopolymers, such as polysaccharides, or their derivatives, such as synthetic poly(amino acids) [4].

Buffington et al., developed an SMP that responds directly to enzymatic activity and can do so under isothermal cell culture conditions. PCL that is vulnerable to enzymatic degradation was used as shape-fixing component and blended with a shape-memory component (Pellethane) that is enzymatically stable; as the shape fixing component underwent enzymatically catalyzed degradation, the SMP returned to its original, programmed shape [49].

### 1.3. Multi-Stimuli Responsive SMPs

Shape memory polymers (SMPs) have been rapidly developed in the last few years and are constantly growing. There is an urgent need to develop smarter systems that can be responsive to different stimuli depending on the situation.

A covalently crosslinked metallosupramolecular polymer exhibited thermo-, photo- and chemo-responsive shape-memory properties. The multiresponsive films obtained consist of a soft poly(butadiene) phase and a hard metal–ligand phase that control fixing and releasing of the temporary shape. Photocrosslinking of the poly(butadiene) soft phase, via the thiol–ene reaction, upon exposure to relatively low-intensity light, allows access to a diverse range of permanent shapes. Moreover, other stimuli such as light, heat or chemicals can disrupt the metal–ligand phase and they can be used to create the temporary shape and induce its recovery back to the permanent shape [50].

Ban et al., reported a novel stimulus-responsive SMP able to deform under UV light and fix shape in visible light; the original shape is then recovered when high temperature is applied [51]. In their work, they defined this particular type of shape-memory behavior as staging-responsiveness.

A poly(vinyl alcohol)-graft-polyurethane comb structure system was developed possessing a multistimuli-responsive effect such as good water-induced and thermal dual shape-memory effect [42].

### 1.4. Multifunctional SMPs

A new class of “intelligent SMPs” was developed not only to achieve SMPs able to respond to different kind of stimuli but also to perform different functions.

Multifunctional SMP materials can be obtained by complexing polymers and/or composites with different functional groups that perform various functions. These functions include degradability, electric conductivity, magnetic conductivity, energy storage, self-healing, antibacterial properties, controlled drug release and so on [7,52].

Using a PCL temperature-responsive degradable polymer for delivery hydrophilic and hydrophobic drugs, Lendlein and coworker obtained a triple-functional polymer network system combining controlled drug release, biodegradation and shape-memory effects. In their study, they demonstrated that: (i) incorporation of hydrophilic and hydrophobic drugs did not influence shape-memory functionality, (ii) a diffusion-controlled release independent from biodegradation rate was enabled, and (iii) the programming and shape recovery process was fulfilled [53].

Liang et al., developed a two-way shape-memory polymer network which was reversible with body temperature and was water responsive [54]. SMP (P60-G40), obtained by random crosslinking of hydrophobic poly(tetrahydrofuran) and hydrophilic poly(ethylene glycol) (PEG) oligomers, was applied as a new type of esophageal stent, and the in vitro assessment showed that the stent was adjustable, self-expandable, and had the ability to load and release drugs.

## 2. Shape-Memory Polymers Nanofibers (SMPNs)

As reported in the previous section, SMPs can be manufactured and obtained with different structures depending on fabrication technique. Concerning SMP fibers, there are many fiber formation processes including melt spinning, dry spinning, wet spinning, gel spinning, emulsion spinning, shear spinning and electrospinning [55].

Among all, electrospinning fibers’ fabrication technique has attracted great attention in recent decades due to its ability to produces fibers ranging from the micrometer to nanometer scale. Electrospinning shows many advantages including easy processing and continuous fibers manufacturing. Electrospun polymer fibers show high surface area per volume unit, high porosity, small diameter, low density, desirable fiber orientation and nanoarchitecture able to mimic native Extra Cellular matrix (ECM). For these reasons, electropsun nanofibers are widely used in different fields as nanosensors, filters, biomedical and tissue engineering supports (i.e., as drug delivery systems, hemostatic gauze, artificial vessels and scaffolds) [56,57]. Moreover, it is reported in the literature that micro/nanofibers show enhanced shape-memory properties compared to shape-memory films [20]. SMP nano/microfibrous membranes, made of the same material and in the same experimental conditions, showed faster shape recovery rate compared to SMP film. This behavior is due to the quicker heating/cooling rate of fibers thanks to their larger surface area.

Electrospinning apparatus is composed of three main parts: syringe needle, high-voltage power supply and collector. Briefly, when an electric field is applied between a needle and a collector, surface charge is induced on the polymer fluid deforming a spherical pendant droplet to a conical shape (Taylor’s cone). As the electric force overcomes the solution surface tension, a polymeric jet is generated from the surface tension of the droplet and travels towards the collector. The solvent evaporates from the jet in the gap between needle and collector, and consequently dry nanofibers are collected [58].

Final fiber morphology and diameter can be controlled by modulating process parameters such as solution parameters (concentration, viscosity, solvent conductivity), process parameters (voltage, flow rate, needle–collector distance, collector types and needle diameter) and environmental parameters (temperature, humidity) [59].

A schematic summary of electrospinning apparatus, working parameters and influence on the fibers properties is reported in Figure 4.

Moreover, implementing electrospinning apparatus and working parameters allows us to obtain better-performing nanofibers. For example, side-by-side spinning of two different polymer solutions allows us to obtain composite fibers, or multifunctional fibers can be achieved by adding functional materials (carbon nanotubes, magnetic nanoparticles, etc.) during the electrospinning process. Finally, core–shell fibers (hollow fibers) are obtained using coaxial spinneret. Due to the advantages above mentioned, SMP nanofibers (SMPNs) have been increasingly investigated [2]. Moreover, fiber shape-memory properties are influenced by fiber diameter and morphology.

Sauter et al., demonstrated that fiber diameter affects macroscopic shape-memory performance of polyetherurethane electrospun meshes. By reducing fibers’ diameter (<100 nm), shape recovery was improved, otherwise, shape fixity was found to decrease with decreasing fiber diameter [60].

Shape changes can be also modulated at the microscopic level; Zhang et al. evaluated and programmed microscopic shape-memory behavior of Nafion (ENNMs blended with 1.0 wt% polyethylene oxide) fibers. Using heating treatment cycles, nonwoven fibers were deformed in an aligned temporary shape and then recovered their original shape [61]. Tseng et al. proved the capability of SMPNs to change their microscopic architecture during cell culture. Starting from a nonwoven electrospun scaffold, alignment was induced to control the behavior of attached and viable cells. The polymeric electrospun nanofibers were obtained using a thermoplastic polyurethane (35–45% *w*/*v*) in a solvent blend of DMF:chloroform = 1:2 (*v*/*v*). Other process parameters set up were a 22 G needle, 0.4 mL/h flow rate and 15.5 kV voltage. A rotating, negatively charged drum (−0.5 kV) with a 10 cm distance from the spinneret to rotating drum surface was used to collect fibers. After 12 h spinning time, scaffold thickness of 100 μm was achieved. They proved that human adipose-derived stem cells preferentially aligned along the fiber direction of the strain-aligned scaffold before shape-memory actuation [62].

Shape memory treatment can be applied also to modulate electrospun fibers’ porosity. Ahn et al. demonstrated that applying heating and cooling cycles above and below polymer T_g_ induced a modification in pore size from 150 to 440 nm. Shape-memory polyurethanes (SMPUs) electrospun membrane (fibers’ dimensions > 1 μm) can be utilized as smart membranes to selectively separate substances according to their sizes by controlling temperature [63]. SMPUs nanofibers studied in their work were obtained using a 4% *w*/*v* concentration, 12 kV voltage, 2 mL/h flow rate, 25 G needle and 10 cm-distance spinneret plate collector.

An electrospun membrane, fabricated by crosslinked PCL and Fe_3_O_4_-loaded MWCNT composite nanoparticles, exhibited excellent magnetic and heat recovery properties. PCL/Fe_3_O_4_ composite nanofibers were obtained using electrospinning equipped with a drum collector (rpm 1000) with a diameter of 16 cm under ultraviolet (UV) irradiation (365 nm-100 W). The voltage applied to obtain homogeneous jet was 15 kV and there was a 20 cm gap between the spinneret and drum collector. After shape-memory treatment, the permanent shape was recovered in 120 s under the stimulation of alternating magnetic field with a frequency of 20 kHz [64].

## 3. Shape-Memory Polymers Nanofibers Biomedical Applications

Tissue-engineered scaffolds have traditionally been static physical structures poorly suited to mimicking the complex dynamic behavior of in vivo microenvironments.

SMPs’ properties can be usefully applied to scaffolds for tissue regeneration to permit minimally invasive surgery implantation which can support cell adhesion and proliferation [20,65].

Shape-memory scaffolds, especially in nanofibers form, exhibit high porosity and specific surface area that have great potentiality for use in tissue engineering. The most common biocompatible and biodegradable SMPs are made from poly(ε-caprolactone) (PCL), polyurethane (PU), poly (D, L-lactide) (PDLLA), PVA, ethylene vinyl acetate copolymer, (EVA) polymer blend, polymer composite, crosslinked polymers and supramolecular networks, with the goal to produce biomedical tissues suitable to mimic the complex dynamic behavior of in vivo microenvironments. Moreover, SMPs able to play an important role in the biological field can become potential in vitro platforms for cell interaction and tissue growth investigations [61].

Therefore, not only are scaffolds and their properties important but also cells and their interaction with scaffolds have an important role in TE.

Mesenchymal stromal cells (MSCs) are a promising tool to treat different pathological conditions, including cancer, autoimmune disorders, degenerative diseases and wound healing [66,67]. MSCs are an outstanding tool for cell therapy applications, not only because of their multipotent nature, but also due to their ability to home and engraft in damaged tissues, release trophic factors, promote neovascularization, manage oxidative stress and trigger anti-inflammatory responses [68,69,70,71,72]. Accumulating evidence suggests that MSCs act through a combination of paracrine cell signaling and cell transdifferentiation, enhancing wound regeneration and improving angiogenesis [73,74]. Recently, it has also been reported that multipotent cells mobilize to the peripheral blood after burn incidents and migrate to the site of injury in response to chemotactic signals where they modulate inflammation, repair damaged tissue and facilitate tissue regeneration [75,76]. The most recognized source of MSCs is bone marrow, however other sources have been described such as adipose tissue, amniotic fluid, umbilical cord and cord blood, teeth, bone, muscle, placenta, liver, and pancreas. Amniotic fluid is an ideal source of stem cells for expansion and banking properties addressed to their therapeutic use [77]. Recently the use of MSCs derived from human amniotic fluid (AF-MSCs) obtained from amniocentesis is increasing due to their advantage as an autologous, multipotent stem cells source. A study performed by Abe et al., demonstrated how AF-MSCs could induce direct coverage of the spinal cord and hepatocyte growth factor secretion [78]. Through an intra-amniotic injection of AF-MSCs in rat model, they showed how the treatment reduced neuronal damage such as neurodegeneration and astrogliosis and promoted neural regeneration.

In Table 2 are reported examples in which thermoresponsive SMPNs were engineered with cells and proposed for cell culture research and tissue regeneration.

Specific applications of SMP to tissue engineering are discussed in the following sub-sections.

### 3.1. Bone Regeneration

Bone tissue engineering was endowed with shape-memory functionality. Besides the capability of enabling minimally invasive surgical implantation, it may also offer the possibility of exerting in situ mechanical stimulus to achieve enhanced efficacy in bone repair and regeneration. Polylactides polymers such as poly(L-lactide) (PLLA), poly(D-lactide) (PDLA), and PDLLA, have been widely used as scaffolding materials for bone regeneration, due to their biocompatibility, tensile strength, and slow degradation rate [84,85,86]. Moreover, polylactide polymers possess an SME if properly programmed. To reach suitable characteristics in terms of toughness and T_g_, polylactides were combined and/or blended with other materials [81].

Bao et al., developed SMPNs bone scaffolds based on poly (D,L-lactide-co-trimethylene carbonate) in different ratios (PLMC 8:2 and 9:1). They also evaluated osteoblast biocompatibility, proliferation and morphology and concluded that the shape-memory fibers could be applied in repairing bone defects, including healing bone screw holes and as barrier membranes for guided bone regeneration [16]. Polyurethane SMPNs were combined with inorganic nanoparticles, such as calcium phosphates and hydroxyapatite (HA), for bone scaffold development. HA nanoparticles provide significant reinforcement of PU. Furthermore, incorporation of HA nanoparticles improved the shape-memory properties of nanocomposite scaffolds [16,87].

A porous smart nanocomposite scaffold made by PCL and HA and loaded with bone morphogenetic protein-2 (BMP-2) showed good cytocompatibility and a suitable shape-memory effect, and was utilized to control growth factors’ delivery and release in order to promote new bone generation in the rabbit mandibular bone defect [42].

Ying et al., developed a HA, collagen and poly(L-lactide-co-caprolactone) (HA/Col/PLCL) composite nanofiber via electrospinning for bone tissue engineering. PLCL/HA/Col scaffolds, characterized for their morphological, structural, thermal–mechanical, shape-memory properties and biological properties, significantly promoted proliferation of rat bone-marrow-derived mesenchymal stem cells (rBMSCs) and enhanced the expression of ALP, Col and mineral deposition [88].

### 3.2. Neural Tube Defects Repair

Neural tube defects (NTDs) represent the second most common cause of congenital malformation in children. NTDs are birth defects of the brain, spine, or spinal cord. NTDs result from abnormal neurulation processes, such as in the process of differentiation from which originates the nervous system [89,90,91]. Myelomeningocele (MMC) is an NTD due to incomplete development of the caudal part of the neural tube [91]. It presents with protrusion of neural tissue and meninx through an opening in the vertebral arches. It is associated with severe morbidity due to lifelong paralysis, bladder incontinence, bowel dysfunction, hydrocephalus, and Arnold Chiari Malformation II. The pathogenesis of the peripheral neurologic defects presents at birth has a double cause, first being the abnormal neural tube development, and secondly, in utero exposure of the open spinal cord to amniotic fluid [90,91].

Tatu et al. developed a self-expanding, watertight and biodegradable patch for fetoscopy MMC prenatal repair [92]. By blending appropriate rates of PLA and PCL they prepared a polymeric film with a glass transition temperature (T_g_) of 37.6 ± 1.28 °C and showing shape-memory behavior in response to external temperature change in the range of the T_g_ value; this property allows unfolding of a coiled scaffold when introduced in the amniotic cavity. In a following paper, the same authors performed an in vivo characterization using a rat model showing that PLA-PCL-blended film did not cause tethering of the cord and did not induce adverse effects on regular functions of the spinal cord [93].

Wang et al., developed an injectable composite hydrogel system for use as a minimally invasive treatment of spinal cord injury using motor neurons derived from embryonic stem cells (ESCs). The hydrogel is composed of a modified gelatin matrix implemented with shape-memory polymer fibers. The gelatin matrix creates a suitable microenvironment for cell assembly, while SMP fibers are able to recover and maintain the microstructures even after drastic deformation due to the injection operation. Moreover, fibers are used as support and guidance for motor neuron differentiation. ESC-loaded composite hydrogel was injected in mice and the authors noticed enhanced tissue regeneration and recovery of motor function [94].

Inspired by the intelligent responsive shaping process of shape-memory polymers, Wang et al., developed a self-forming, multichannel, nerve guidance conduit with topographical cues using degradable shape-memory PLA-trimethylene carbonate copolymer. An electrospun shape-memory nanofibrous mat could be temporarily formed into a planar shape for cell loading with uniform distribution. Then, triggered by a physical temperature around 37 °C, the electrospun shape-memory nanofibrous mat could automatically restore its permanent tubular shape to form the multichannel conduit. This study demonstrated that the fabricated bioinspired multichannel nerve guidance conduit has great potential in peripheral nerve regeneration [83].

Engineered shape-memory electrospun scaffolds (E-SMESs) were developed by our group using a PLA-PCL copolymer (70:30 ratio) to produce nanofibers that possess a thermoresponsive behavior. Electrospun scaffolds were cellularized with amniotic fluid mesenchymal stem cells and intended as scaffold for neural tube defects’ repair. E-SMES were rolled up and then underwent a cycle of high (T° > T_g_°) and low temperature (T° < T_g_°) in order to induce copolymer solid status change from rubbery to glassy and to fix their rolled temporary shape. The scaffolds were then immersed in a PBS bath at 37 °C (body physiological temperature) and returned to the original flat configuration (Figure 5) [95].

### 3.3. Vascular Graft

Cardiovascular diseases are the leading cause of death worldwide. Artificial vascular graft could be a solution for treating vascular disorders and endothelial cell dysfunction. However, a rapid formation of a confluent endothelial monolayer onto the lumen of 3D structure is essential in order to achieve a suitable regenerative process [96]. To reach this goal, novel shape-morphing scaffolds were developed, enabling programmed deformation from planar shapes to small-diameter tubular shapes, and were designed combining a biocompatible shape-memory polymer with an electrospun nanofibrous membrane [97].

Zhao et al., fabricated a membrane of nanofibrous PCL and gelatin methacrylate able to readily convert its 2D planar shapes into 3D tubular shapes for facile 3D endothelialization [98]. As a proof of concept, human umbilical vein endothelial cells (HUVECs) were randomly seeded onto the shape-morphing scaffolds in their temporary planar shapes at room temperature. After putting the engineered shape-morphing scaffolds into a 37 °C incubator for cell culture, the planar scaffolds were able to recover their permanent 3D tubular shapes with well-preserved cell viability and desirable cell attachment. The shape-morphing scaffolds developed are a great promise for promoting small-diameter vascular tissue engineering.

A new library of property-tunable SMPs was synthesized by Shin et al. through ring-opening polymerization with ε-caprolactone and glycidyl methacrylate monomers [99]. The performance of vascular grafts obtained (diameter less than 5 mm) was examined in a porcine model, and it was confirmed that the graft was able to prevent vessel damage by expanding the graft diameter circumferentially upon implantation. Moreover, the graft coating, with nitric-oxide-releasing peptides, minimizes the disturbed flow formation and prevents thrombosis.

### 3.4. Skin Wound Healing

Skin plays a crucial role in different processes such as hydration, protection from chemicals and pathogens, vitamin D synthesis initialization, excretion and thermal regulation. For these reasons, severe skin damage can be life-threatening. The skin repair process includes the interaction of cells, growth factors and cytokines involved in closing the lesion [100]. Shape-memory property can endow polymer wound dressings with the performance of maintaining their original shape under external force, and can benefit the initial stage of wound healing through shape recovery-assisted closure of cracked wounds. Therefore, wound dressings with shape-memory property have great potential to enhance the skin wound healing process [101,102].

Tan et al. prepared a series of GO-filled SMPU nanofibers; the presence of GO significantly improves the mechanical strength, surface wettability, and thermal stability of the SMPU for a wound healing application [103].

Li et al., designed and synthesized a series of electroactive shape-memory polyurethane-urea elastomers by combining the mechanical properties of PCL, wettability of polyethylene glycol (PEG) segments and electroactivity of aniline trimer, as an antibacterial, antioxidant and electroactive film dressing for cutaneous wound healing. Moreover, after loading with vancomycin as an antibacterial agent, the rate of wound healing was further enhanced [104].

## 4. Sterilization and Shape Memory Effect

Important requirements of SMPs-scaffolds for biomedical applications are biocompatibility, to guarantee good cell viability and material interaction, and biodegradability, to avoid a second surgical operation to remove the implanted materials. Sterility is another unavoidable requirement of scaffolds intended to be implanted into the human body, thus, it needs be taken into account during the development of SMPs for in vivo biomedical application. Sterilization processes must guarantee scaffolds’ structural and biochemical properties while maintaining their functionality poststerilization [105].

The choice of the most appropriate technique is based on the type of SMP and on the stimulus needed to regain scaffold shape, considering that sterilizing treatments can modify scaffolds’ characteristics. Sterilization by heat, either dry or steam heat, can be harmful for almost all polymer-based products since heat can induce polymer degradation or unwanted crosslinking. Moreover, thermally sensitive SMPs cannot be sterilized by heat application because SME can be altered. The most commonly used sterilization techniques are ethylene oxide (EtO) and gamma irradiation.

Ethylene oxide belongs to the category of chemical sterilization technique and acts by causing irreversible alkylation of cellular molecules that may contain amino, carboxyl, thiol, hydroxyl, and amide groups, resulting in permanent suppression of cell metabolism and division. Unfortunately, one of the main disadvantages of this sterilization technique is the residual toxicity of EtO that could remain in the scaffolds. The American National Institute for Occupational Safety and Health (NIOSH) have set guidelines of 25–250 ppm as the maximum EtO residual concentration in medical devices post-EtO sterilization, with a recommended range of 10–25 ppm [106].

Gamma irradiation is an ionizing radiation technique usually achieved through a source of 60 Co and produced within a dose range of 10–30 kGy/h. The radiation dose recommended by the European Pharmacopoeia is 25 kGy, to ensure that the Sterility Assurance Level (SAL) of 10^−6^ is respected, but lower radiation doses can be used provided the process has been validated demonstrating 10^−6^ SAL achievement [107].

Gamma irradiation is advantageous for heat-sensitive polymeric materials because it operates at low temperatures, in short times. However, polymers can undergo chemical, mechanical, and morphological changes, such as polymer degradation by chain scission or polymer cross-linking, or both. Cleavage is observed at the level of weak bonds with breakage and subsequent decrease in molecular weight. Cross-linking leads to the formation of large three-dimensional networks that are fragile and prone to degradation. For example, PLA may undergo morphological changes after gamma ray treatment such as the appearance of rougher surfaces due to cleavage. Mechanical changes such as an increase in yield point can be seen in PCL after irradiation [105,108].

Molecular weight reduction due to cleavage at the level of the polymer chains leads to an increase in polymer chains’ mobility, with a reduction in T_g_ values [109].

Rychter et al. evaluated the effect of gamma irradiation (in the range from 10 to 25 kGy) for sterilization of porous scaffolds with shape-memory behavior obtained from biodegradable terpolymers: poly(l-lactide-co-glycolide-co-trimethylene carbonate) and poly(l-lactide-co-glycolide-co-ɛ-caprolactone) [110]. Treatment of the samples with gamma irradiation at 15 kGy resulted in a considerable drop in glass transition temperature (T_g_), decrease in average molecular weight (M_n_) and in scaffolds’ mechanical properties. However, sterilization did not influence the shape-memory behavior of the examined materials.

Bosworth et al., reported the effects of gamma irradiation (range between 0–45 kGy) on electrospun PCL fibers [111]. A dose-dependent reduction in molecular weight was reported and also an increase in melting point and crystallinity. PCL fibers exposed to a higher irradiation dose (40 kGy) also reported a decrease in tensile strength. However, sterilization did not affect cell response.

The reported studies demonstrate that is important: (i) to follow good manufacturing practice during scaffold preparation in order to limit microbiological contamination, and (ii) to include a medical-grade sterilization procedure in order to assist translation from bench to clinic.

## 5. Conclusions and Future Prospective

It is evident that SMPs have achieved great progress in architecture, triggering new methods and biomedical applications. Extensive work has been conducted to identify network design principles that can tune polymer chemistries for a variety of applications, with increasing effort towards understanding SME and biological response to these materials. Particularly, SMPNs attract lot attention due to their faster stimulus response, good flexibility, low weight, easy processing and modulating structures. Especially in biomedical field, SMPs, in micro/nanofiber form, showed a dynamic fibrous structure suitable to guide cell regulation and differentiation, control drug delivery, and guarantee scaffolding.

However, several challenges still remain to be addressed and improved. Considering different shape-memory effects, TWSME is to be preferred over OWSME, so reversible micro/nanostructure SMPs should be further investigated to achieve better performing and more durable scaffolds. Thermoresponsive SMP fibers are the most investigated due to their wide applications, but it is still necessary to program a more accurate transition temperature range for particular biomedical applications; in fact, different temperatures are registered during inflammation, infection or other pathological conditions. Stimulation using indirect triggers is preferred in many fields, and the development of SMPs combined with functional particles and responding to multiple stimuli is to be considered when applying SMPs in different pathological conditions or in sites with difficult access.

Moving toward personalized medicine and customized devices, additional innovative manufacturing technologies can be used to produce SMP constructs with desired properties. 3D printable SMPs are well-positioned to address the needs of various clinical challenges requiring patient-specific design device customization. Dynamic SMP fibrous scaffolds were used in vitro to study cell behavior, recovery processes, degradation and tissue formation. However, it is still a challenge to evaluate their effect in vivo, for example in bone, neural tube and skin tissue regeneration. Finally, an extensive preclinical investigation is required to translate SMPs and SMPNs from laboratories to clinical applications.

## Figures and Tables

**Figure 1 ijms-23-01290-f001:**
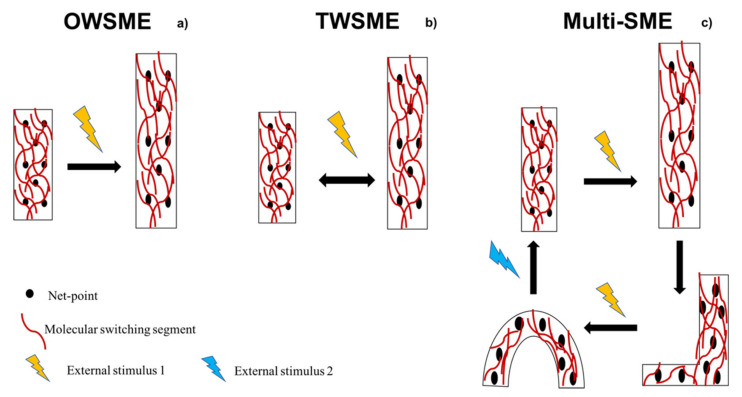
SMPs with different shape-memory effect (SME): (**a**) one-way (OWSME), (**b**) two-way reversible (TWSME) and (**c**) multiple-SME.

**Figure 2 ijms-23-01290-f002:**
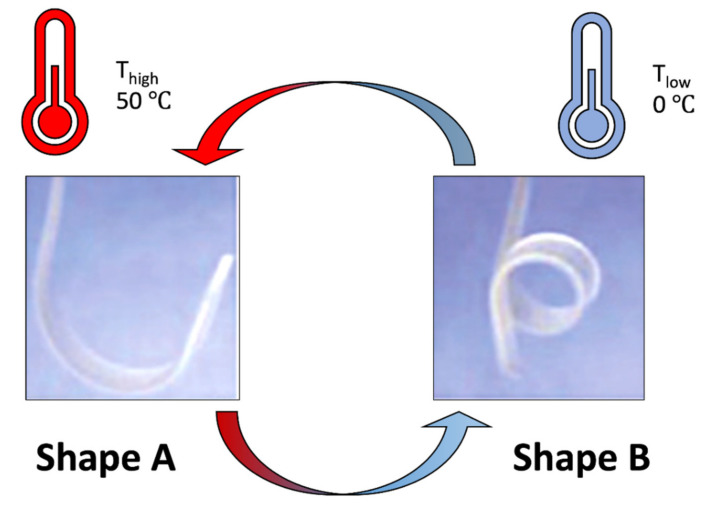
Photograph series showing reversible bidirectional SMPs (40 mm × 4 mm × 0.4 mm) from PPD-PCL. The bowed shape was obtained after programming by deformation in a helixlike shape at T_reset_, cooling to T_low_ and subsequent heating to T_high_. The SME occurred as reversible shift between shape A (bow) at T_high_ and shape B (helix) at T_low_. Image was modified from Behl et al. paper [22].

**Figure 3 ijms-23-01290-f003:**
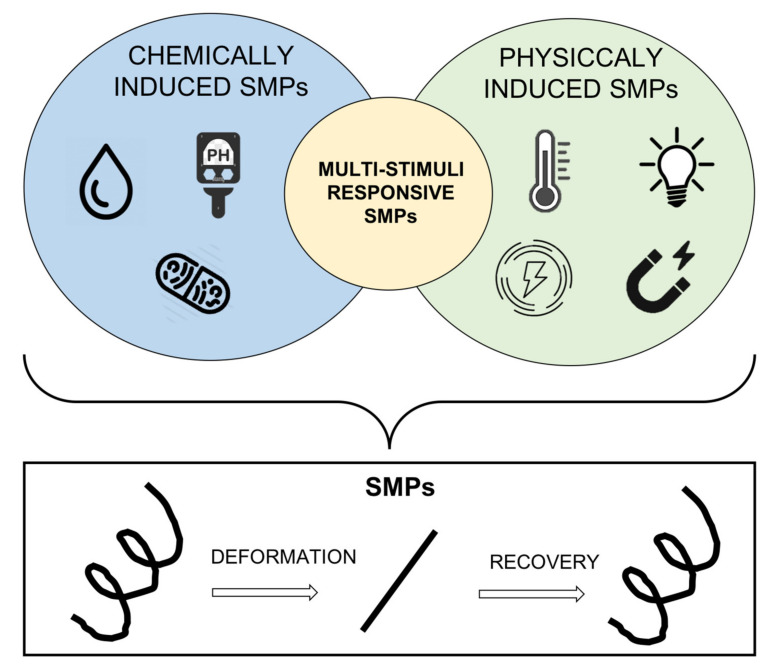
Scheme of different stimuli-induced SMPs.

**Figure 4 ijms-23-01290-f004:**
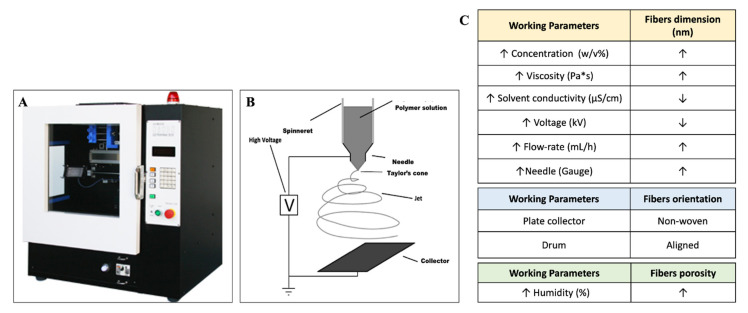
(**A**) Electrospinning NANON-01A; (**B**) Scheme of vertical electrospinning apparatus composed by syringe needle, high-voltage power supply and collector; (**C**) working parameters and their influence on fibers properties; the arrow symbol states for increase; the arrow symbol states for decrease.

**Figure 5 ijms-23-01290-f005:**
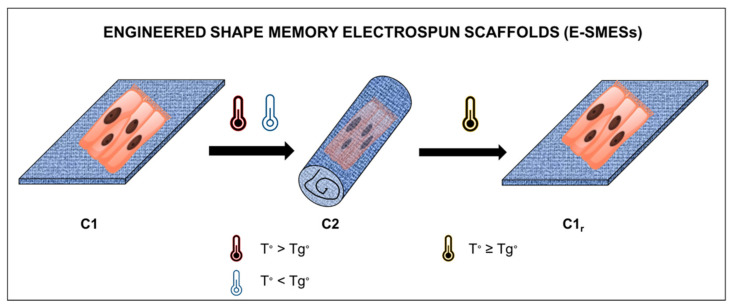
Engineered shape-memory electrospun scaffolds (E-SMESs). C1 original flat conformation; C2 temporary rolled-up conformation; C1r recovered original flat conformation.

**Table 1 ijms-23-01290-t001:** Biodegradable Shape-Memory Polymers (BSMPs) and their application in biomedical fields.

BSMPs	Biomedical Application	Ref.
PCL *	Tracheal stent	[5,6]
	Drug release	[7]
	Sutures	[8]
PU	Embolization	[9]
	Contraception	[10]
	3D scaffolds	[11]
	Hemostatic devices	[12]
PLA	Stent	[13,14]
	Bone tissue engineering	[15,16]
PLGA	Embolization	[17]
	3D scaffolds	[18,19]

* PCL and crosslinked forms.

**Table 2 ijms-23-01290-t002:** Examples of cell lines combined to shape memory polymer (SMPNs) nanofibers.

SMPNs	Cell Line	Purposes	Ref
Poly(lactide–glycolide)/chitosan	Smooth muscle cells	Regulating cell adhesion, proliferation, and morphology	[79]
Poly(ε-caprolactone) with hexamethylene diisocyanate/1,4-butanediol	Human mesenchymal stem cell	Altering human mesenchymal stem cell alignment and orientation (cell culture platform)	[80]
Poly(3-Hydroxybutyrate-co-3-Hydroxyvalerate) Modified Poly(l-Lactide)	Mouse bone mesenchymal stem cells	Enhanced osteogenesis-inducing ability in bone mesenchymal stem cells for applications in bone tissue repair and regeneration.	[81]
Poly-L-lactide-co-poly-ε-caprolactone	Mesenchymal stem cell	Engineered shape-memory electrospun scaffold to promote neural tube defects’ repair	[S. Pisani et al., submitted for publication in Journal of Reactive and Functional Polymers]
Poly-DL-lactic acid-based polyurethane	Human fibrosarcoma cell line HT-1080	On-command on/off switching of cell polarized motility and alignment	[82]
Poly(lactide-co-trimethylene carbonate)	Schwann cells (SCs)	Multichannel nerve guidance conduit for potential application in peripheral nerve repair	[83]

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
