# Peer review of "Shape-Memory Polymers Hallmarks and Their Biomedical Applications in the Form of Nanofibers"

_ijms, 2022, doi:10.3390/ijms23031290_

Round 1

Reviewer 1 Report

This review article ''Shape memory polymers hallmarks and their biomedical applications in the form of nanofibers'' is well written and informative.
In my opinion the paper is acceptable in its current form. Therefore, I would recommend to accept it for publication in IJMS.

Author Response

Review comment: This review article ''Shape memory polymers hallmarks and their biomedical applications in the form of nanofibers'' is well written and informative.
In my opinion the paper is acceptable in its current form. Therefore, I would recommend to accept it for publication in IJMS.

Authors reply: We  thank the reviewer for its comment and for appreciating our work.

Reviewer 2 Report

The manuscript  'Shape Memory Polymers Hallmarks and Their Biomedical Applications in the Form of Nanofibers' represents an actual review of shape memory polymers and nanofibers therefrom.

In general, the subject of the review meets the high level of IJMS by the criteria of scientific relevance. However, the structure of the Section 1 of the manuscript – solely in my opinion – is very similar to the recent review [Chin. J. Polym. Sci. 36 (2018) 905. doi: 10.1007/s10118-018-2118-7] and therefore the article needs for additional editing and expansion. For the further improvement of the manuscript, please, take in mind recommendations below.

Section 1.

First, it would be appropriate to refer on previous review  [Adv. Healthcare Mater. 6 (2017) 1700694]. It seems to me that the manuscript need addition of the chemical structures of copolymers and functionalized copolymers that demonstrate shape memory effect.

Also, a number of references deserve citation [Polym. Chem. 6 (2015) 4177; Macromolecules 50 (2017) 8570; Polymer 83 (2016) 40; Eur. Polym. J. 48 (2012) 1866; Polymer 55 (2014) 5953; Adv. Mater. 22 (2010) 3424; Adv. Funct. Mater. 20 (2010) 3583; J. Mater. Sci.: Mater. Med.  22 (2011) 2147;  Macromol. Mater. Eng. 297 (2012) 1184; ACS Macro Lett. 8 (2019) 682; Macromol. Chem. Phys. 219 (2018) 1700345].

Section 2.

ES technique is well-known. It would be more appropriate to specify preparation of SMPNs of different types using ES.

Section 3.

No substantive comments.

Section 4.

New and important, no substantive comments.

Minor remarks

Line 16 – replace 'nanfiber' by 'nanofiber'

Line 43 – replace ' classified' by 'classify' or 'categorize'

Line 45 – not 'in' but 'on' or 'as'

Line 48 and below – excess of the usage of indents

Lines 52–59 – maybe the use of bullet points is more visible?

Line 69 and below – Tm, 'T' italic, 'm' subscript; throughout the text T (temperature) – italic, as variable

Line 70 – replace 'peptidolactone' by 'pentadecalactone'

Line 92 – replace 'is possibleto' by 'it is possible to'

Line 115 – please be careful and eliminate editing remarks

Lines 140-141 – please add the references on patents mentioned in the text

Line 182 – replace 'Chemical' by 'Chemically'

Line 213 – replace 'poly vinyl acetate' by 'poly(vinyl acetate)'

Line 268 – replace 'polytetrahydrofuranand' by 'poly(tetrahydrofuran) and'

Line 303 – please improve the quality of the Fig. 4

Line 346 – replace 'ethylenevinyl acetate' by 'ethylene – vinyl acetate'

Line 354 – 'to treat different conditions' – what do you mean?

Line 383 – the references on the use of PLLA, PDLLA and especially poly(D-lactide) are needed

Line 439 – replace 'PLA- trimethylene carbonate olymer' by 'PLA – trimethylene carbonate copolymer'

Lines 446-456 – the reference is needed

Line 468 – replace 'intheir' by 'in their'

etc, etc.

Author Response

Author answer to reviewer 2  have been uploaded here below
